# Effect of Isolation Conditions on Diversity of Endolichenic Fungal Communities from a Foliose Lichen, *Parmotrema tinctorum*

**DOI:** 10.3390/jof7050335

**Published:** 2021-04-26

**Authors:** Ji Ho Yang, Seung-Yoon Oh, Wonyong Kim, Jung-Jae Woo, Hyeonjae Kim, Jae-Seoun Hur

**Affiliations:** 1Korean Lichen Research Institute, Sunchon National University, 255 Jungang-Ro, Suncheon 57922, Korea; 836019@naver.com (J.H.Y.); apbiomol@gmail.com (W.K.); wjdwoz@hanmail.net (J.-J.W.); khj4173@hanmail.net (H.K.); 2Department of Biology and Chemistry, Changwon National University, 20 Changwondaehak-ro, Changwon 51140, Korea; syoh@changwon.ac.kr

**Keywords:** diversity, endolichenic fungi, foliose lichen, isolation method

## Abstract

Endolichenic fungi (ELF) are emerging novel bioresources because their diverse secondary metabolites have a wide range of biological activities. Metagenomic analysis of lichen thalli demonstrated that the conventional isolation method of ELF covers a very limited range of ELF, and the development of an advanced isolation method is needed. The influence of four variables were investigated in this study to determine the suitable conditions for the isolation of more diverse ELF from a radially growing foliose lichen, *Parmotrema tinctorum.* Four variables were tested: age of the thallus, severity of surface-sterilization of the thallus, size of a thallus fragment for the inoculation, and nutrient requirement. In total, 104 species (1885 strains) of ELF were isolated from the five individual thalli of *P. tinctorum* collected at five different places. Most of the ELF isolates belong to Sordariomycetes. Because each part of lichen thallus (of different age) has unique ELF species, the whole thallus of the foliose lichen is needed to isolate diverse ELF. Moderate sterilization is appropriate for the isolation of diverse ELF. Inoculation of small fragment (1 mm^2^) of lichen thallus resulted in the isolation of highest diversity of ELF species compared to larger fragments (100 and 25 mm^2^). Moreover, ELF species isolated from the small thallus fragments covered all ELF taxa detected from the medium and the large fragments in this study. The use of two media—Bold’s basal medium (nutrient poor) and potato dextrose agar (nutrient rich)—supported the isolation of diverse ELF. Among the tested variables, size of thallus fragment more significantly influenced the isolation of diverse ELF than other three factors. Species composition and richness of ELF communities from different lichen thalli differed from each other in this study.

## 1. Introduction

Lichens result from a symbiotic association between fungi and their photosynthetic partners [1], in which diverse microbial communities, including the endolichenic fungi (ELF), are housed. The ELF live inside the lichen thalli without any disease symptoms and are similar on their lifestyle to endophytic fungi [2,3]. They depend on photosynthesizing partner of lichen for their carbon source as heterotroph [4]. Therefore, ELF within the thallus of lichen are known to be abundantly located at the algal layer [3]. Taxonomic composition of ELF communities is distinct from that of endophytic fungal communities [5,6]. Several factors including type of host lichen, climate, and geographical region affect construction of ELF community [5,6,7,8]. Various ELF have a range of biological activities owing to their unique secondary metabolites. Novel natural products derived from ELF showed diverse bioactivities including antibacterial, antifungal, antioxidative, and cytotoxic effects [9,10,11,12]. Therefore, ELF are considered as novel and potential biological resources for pharmaceutical and biotechnological purposes.

Metabarcoding results showed that there are highly diverse fungi in the lichen thallus [13]. Several studies have been performed in order to reveal the influence of different isolation conditions on the composition of culturable ELF communities. Muggia et al. reported the effects of different media on the community of isolated ELF [14]. Previous studies showed that the composition of isolated ELF was affected by different methods of sterilization of lichen surface [15,16]. Most of the studies focused only on the testing of a single factor of isolation conditions but not on their combined ability to increase the diversity of ELF. Therefore, more effort is needed to improve the conventional isolation conditions to explore novel ELF.

In this study, four isolation variables—thallus age, severity of surface sterilization, inoculum size, and type of medium—were tested to optimize isolation conditions ensuring the isolation of diverse ELF communities from a foliose lichen, *Parmotrema tinctorum*.

## 2. Materials and Methods

### 2.1. Lichen Sampling

*Parmotrema tinctorum* (Despr. ex Nyl.) Hale., a radially growing foliose macrolichen, was used as a model lichen species for the following reasons: it is distributed widely in the tropical and subtropical regions of the world [17,18]; it is loosely attached to the substrate so that the thallus can be peeled off easily and cleanly from the substrate, e.g., bark and rock; mature thallus of *P. tinctorum* has sufficient biomass for the test. Five individual lichen thalli were collected from the bark of pine trees (*Pinus thunbergii* Parl.) at five different sites of Jeju Island located at the subtropical zone of the southern part of South Korea. The climate of Jeju Island is affected by Southeast Asian monsoon [19]. Heavy rainfall derived from the East China Sea and the West Pacific contributes to high levels of humidity in summer. The climate during winter is characterized by cold and dry conditions. The mean annual temperature is around 25.4 and 5.1 °C in summer and winter, respectively [20]. Five collection sites were the following: site 1 (33°28′31″ N, 126°21′09″ E) is located at a seaside cliff and directly exposed to strong winds from the sea; site 2 (33°30′28″ N, 126°28′01″ E) is a hill near the sea; site 3 (33°32′59″ N, 126°45′26″ E) is located at the top of the rising small defunct volcano which has been used as a graveyard and surrounded by agricultural fields (citrus farms); site 4 (33°14′1″ N, 126°22′59″ E) is located at a small park nearby the sea; and site 5 (33°16′23″ N, 126°42′14″ E) is located along the seaside with densely-developed viridian forest (Figure 1a).

### 2.2. Isolation of ELF: Modification of Four Variables

ELF isolation was conducted based on the conventional methodology Appendix A [5,21]. The residue (e.g., bark particles and web) on the lichen surface was removed physically using a syringe tip and running tap water. Lichen thallus was sterilized with ethanol and sodium hypochlorite (NaOCl) and then cut to the required size (e.g., 0.5 cm^2^) [15,21]. Finally, the sterilized thallus fragments were inoculated on a nutrient medium. Modification of the isolation conditions was attempted as follows. Each lichen thallus was divided into three parts (center, middle, and margin) by the diameter of thallus from center according to the age (older, old, and young) (Figure 1b). The lichen thallus was treated by different means of surface sterilization: 70% ethanol, followed by 0.4% NaOCl, during 60, 90, and 120 s (hereinafter referred to as mild, moderate, and severe, respectively), and then rinsed with sterilized distilled water (SDW) for 120 s. The rinsed SDW was smeared on potato dextrose agar (PDA) to confirm the complete sterilization of the thallus surfaces. Sterilized thallus of three different parts was cut carefully into three different sizes of 100, 25, and 1 mm^2^ (hereinafter referred to as large, medium, and small, respectively) (Figure 1c). Large and medium thallus fragments were plated in 90-mm Petri dishes, four and six fragments per plate, respectively. Ten plates were inoculated with fragments of each size; altogether, 40 and 60 fragments of large and medium size, respectively, were incubated. Due to shortage of lichen thalli, only 2 thalli of *P. tinctorum* from site 3 and 5 were used to prepare large and medium thallus fragments. Four 24-well plates were inoculated with 24 pieces of the small fragments; altogether 96 small fragments were tested. The thallus fragments were inoculated on a media consisting of PDA, malt and yeast extract agar (MY, Difco, Spark, MD, USA) [22], Lysogeny broth (LB, Difco, Spark, MD, USA) [23], and Bold’s basal medium (BBM) [24]. The fragments of medium size from the middle part of thallus were incubated on PDA after moderate sterilization as a control. The plates were incubated at room temperature (RT, ca. 21.5 °C) for more than 2 months. Newly growing mycelium was transferred to a fresh PDA for the pure culture. Pure isolates were incubated at 20 °C until sufficient biomass for DNA extraction was obtained.

### 2.3. Molecular Identification

The isolates were grouped into different morphotypes based on their morphological characteristics [25]. The genomic DNA (gDNA) of each morphotype was extracted using a PowerSoil DNA Isolation Kit (Qiagen, Hilden, Germany). The nuclear ribosomal internal transcribed spacer (ITS) region was amplified using the primers, ITS4 (TCCTCCGCTTATTGATATGC) and ITS5 (GGAAGTAAAAGTCGTAACAAGG) [26]. A polymerase chain reaction (PCR) was performed using PCR PreMix (Bioneer, Daejeon, Korea) in a final volume of 20 μL containing 1 μL of 10 pmol of bidirectional primer, 2 μL of gDNA, and 16 μL of SDW. The PCR conditions were as follows: 95 °C for 5 min, and 30 cycles of 94 °C for 30 s, 56 °C for 30 s, 72 °C for 60 s, and a final extension at 72 °C for 5 min. The PCR product was checked using 1% agarose gel electrophoresis. Sequencing was accomplished using the GenoTech (Daejeon, Korea). The bidirectional sequences were edited, quality-checked using BioEdit ver. 7.0.5.3 (http://www.mbio.ncsu.edu/BioEdit/bioedit.html, 1 March 2008) [27] and assembled using ATGC ver. 1.03 (GENETYX co, Tokyo, Japan). The whole sequences in this study were deposited in GenBank (Appendix A). ITS sequence-based identification was conducted using BLAST against NCBI GenBank. Phylogenetic analysis was also conducted using the software MEGA ver. 7.0 (USA) [28] in the maximum-likelihood (ML) at the species level.

### 2.4. Statistics and Visualization

The phylogenetic trees were annotated and edited using the web-based tool, iTOL [29]. All statistical analyses were performed in R ver. 3.5.1 [30]. The rarefaction curve was constructed using the iNEXT package [31]. The isolate density was calculated by dividing the number of isolates by the number of segments of lichen thallus used: *n*/40 (large), *n*/60 (medium), and *n*/96 (small) using the vegan package [32] and visualized using the ggplot2 package [33]. Isolate densities and species richness were compared using a Kruskal–Wallis test [34] followed by a Bonferroni correction [35] (multiple comparisons) and a Wilcoxon matched pairs test [36] (pairwise comparisons). Construction of the Venn diagrams was performed using the VennDiagram package [37]. Non-metric multidimensional scaling (NMDS) was set up using the vegan package based on the Bray–Curtis matrix [38].

## 3. Results

### 3.1. Composition of ELF Biota

Phylogenetic analysis showed that all the host samples belong to *P. tinctorum* (Appendix A). In total, 1885 strains were recovered from 5 thalli of *P. tinctorum* collected at different sites. As the depth of isolation almost reached the saturation point of the rarefaction curve, the sampling effort was sufficient to obtain the diverse species richness of the isolated ELF (Figure 2). After phylogenetic analysis, all the isolates were classified into 104 species, representing 2 phyla, 7 classes, 23 orders, 38 families, and 60 genera (Figure 3, Appendix A and Appendix A).

Ascomycota was the dominant phylum (1736 isolates, 92.10%). In Ascomycota, the most abundant class was Sordariomycetes (1618 isolates, 85.84%), followed by Eurotiomycetes (73 isolates, 3.87%) and Dothideomycetes (37 isolates, 1.96%). At the order level, Xylariales (846 isolates, 44.88%) was dominant, followed by Hypocreales (479 isolates, 25.41%) and Trichosphaeriales (109 isolates, 5.78%). At the family level, Xylariaceae (812 isolates, 43.08%), Sarocladiaceae (248 isolates, 13.16%), and Hypocreaceae (203 isolates, 10.77%) were abundant. At the genus level, *Daldinia* (278 isolates, 14.75%) was dominant, followed by *Sarocladium* (248 isolates, 13.16%) and *Trichoderma* (203 isolates, 10.77%). The most frequently isolated species was the *Daldinia childiae* complex (276 isolates, 14.64%), followed by *Sarocladium kiliense* (248 isolates, 13.16%) and *Trichoderma* sp. (152 isolates, 8.06%). In Basidiomycota, 17 species were isolated. The most abundant species was *Bjerkandera adusta* (Phanerochaetaceae, Polyporales, 30 isolates, 1.59%) followed by *Irpex lacteus* (Irpicaceae, Polyporales, 18 isolates, 0.95%), *Hymenochaete yasudai* (Hymenochaetaceae, Hymenochaetales, 14 isolates, 0.74%), and *Trametes versicolor* (Coriolaceae, Polyporales, 14 isolates, 0.74%). The most abundant species, *Daldinia childiae* complex, was isolated from all study sites and under different isolation conditions (Figure 3). Some of the ELF belonging to Sordariomycetes (e.g., *Sarocladium kiliense*, *Trichoderma* sp., *Coniochaeta veutina*, Microasclaes spp., *Nigrospora* spp. and several species of Xylariaceae) were also isolated regardless of the isolation conditions. Several taxa were isolated only under specific isolation conditions.

### 3.2. Effect of Isolation Conditions on Compositions of ELF Communities

#### 3.2.1. Effect of Size of Thallus Fragment on ELF Diversity

The sizes of thallus fragment remarkably affected diversity of ELF communities. Isolate density (ID) and species richness were significantly different among the three different sizes of thallus fragment (Figure 4a,b). IDs of ELF communities from small, medium, and large part were 0.73 ± 0.22, 0.47 ± 0.02, and 0.66 ± 0.06, respectively (*n* = 2). The smaller the fragments that were used, the more diverse ELF were isolated. The highest species richness was observed in ELF community from small segment (27.5 ± 1.5, *n* = 2), followed by that from medium (8.5 ± 0.5, *n* = 2) and large segments (7 ± 1, *n* = 2). Almost 4-times higher number of ELF species were recovered from small thallus fragment, compared to large thallus fragment (Figure 4b). As shown in Figure 5a, use of small thallus fragment (1 mm^2^) covered all ELF taxa isolated from medium (25 mm^2^) and large (100 mm^2^) fragments. Notably, much more diverse ELF representing five classes (Agaricomycetes, Dothideomycetes, Eurotiomycetes, Pezizomycetes, and Sordariomycetes) were isolated from the small inoculum. Several orders of Sordariomycetes were isolated only from the medium and large inoculum (Appendix A).

#### 3.2.2. Effect of Thallus Age on the ELF Diversity

Because the use of small thallus fragments gave the possibility to isolate all ELF taxa detected in the large and medium thallus fragments, we used small thallus fragment to evaluate effects of other isolation conditions on compositions of ELF communities. The ELF communities from different part of thallus representing different age showed a significant difference in ID (isolate density) (Kruskal–Wallis, *p* < 0.05) (Figure 4c). The ELF communities from the old thallus of the middle part (0.68 ± 0.016, *n* = 5) and the center part (0.51 ± 0.026, *n* = 5) showed significantly higher IDs than that of the young thallus of the marginal part (0.36 ± 0.13, *n* = 5) (Wilcoxon, *p* < 0.05). Species richness of the ELF communities isolated from different thallus parts were also significantly different (Kruskal–Wallis, *p* < 0.05) (Figure 4d). The highest Richness was observed in the middle part of the lichen thallus (23.80 ± 4.32, *n =* 5) followed by the center (14.40 ± 4.62, *n =* 5) (Wilcoxon, *p* < 0.05), and marginal parts (12.40 ± 2.61, *n =* 5) (Wilcoxon, *p* < 0.05). From the thallus fragments of all tested sizes, 70, 43, and 39 species were isolated from the middle, center, and marginal parts, respectively (Figure 5b). Notably, several specific taxa were isolated exclusively from each part of the lichen thallus. For example, *Peniophora crassitunicata*, *Phlebia acerina*, *Paraconiothyrium brasiliense*, Dothideomycetes sp., *Penicillium* aff. *stecki*, Sordariales sp. and *Cercophora caudata* were only isolated from the middle part. *Cerrrena zonata, Cineromyces lindbladii*, *Hypoxylon howeanum*, *Lenzites betulinis*, *Nemania plumbea*, *Rosellinia limonispora*, and *Xylaria* sp. 1 were detected only in the center part. *Coniochaeta boothii*, *Mollisia* sp., *Muscodor fengyangensis*, *Trametes cubensis*, *Virgaria nigra*, and Xylariaceae sp. 3 were solely isolated from the marginal part (Figure 3).

#### 3.2.3. Effect of Severity of Thallus Surface Sterilization on ELF Diversity

IDs of ELF communities isolated after different regimes of surface sterilization differed significantly (Kruskal–Wallis, *p* < 0.05) (Figure 4e). Severe sterilization significantly reduced the ID (0.29 ± 0.05, *n* = 5) compared to mild (0.63 ± 0.064, *n =* 5) and moderate sterilization (0.68 ± 0.17, *n* = 5) (Wilcoxon, *p* < 0.05). The Richness was also significantly different after the three sterilization regimes. (Kruskal–Wallis, *p* < 0.01) (Figure 4f). Moderate sterilization (23.8 ± 4.32, *n* = 5) showed the highest Richness followed by mild (14 ± 2.34, *n* = 5) (Wilcoxon, *p* < 0.05) and severe sterilization (9 ± 1.58, *n* = 5) (Wilcoxon, *p* < 0.01). In particular, ELF belonging to Dothideomycetes were not isolated from the severely sterilized thallus (Appendix A). From thallus fragments of all tested sizes, 47, 70 and 29 species were isolated after mild, moderate and severe sterilization of 5 lichen thalli, respectively (Figure 5c). The growth of fungal hyphae was detected on PDA several days after inoculation of the rinsed SDW of the mild-sterilized thallus. On the other hand, no fungal hyphae growth was noted in the rinsed SDW of the moderate- and severe-sterilized thallus.

#### 3.2.4. Effect of the Medium Type on the ELF Diversity

The IDs of ELF communities isolated on different media were remarkably different (Kruskal–Wallis, *p* < 0.01) (Figure 4g). The IDs on BBM, LB, MY and PDA were 0.38 (±0.04, *n =* 5), 0.3 (±0.07, *n =* 5), 0.55 (±0.04, *n =* 5) and 0.68 (±0.17, *n =* 5), respectively. The IDs on PDA and MY were significantly higher than those on LB and BBM. The species richness of ELF on PDA (23.8 ± 4.32, *n* = 5) was much higher than those on BBM (15.2 ± 3.90, *n* = 5) Wilcoxon, *p* < 0.05), LB (9.2 ± 1.79, *n* = 5) (Wilcoxon, *p* < 0.05), and MY (12.4 ± 2.50, *n* = 5) (Wilcoxon, *p* < 0.05) (Figure 4h). From thallus fragments of all tested sizes, 70, 54, 42 and 29 ELF species were recovered from PDA, BBM, MY and LB, respectively (Figure 5d). In particular, higher number of isolates of slow-growing fungi belonging to Chaetothyriales were recovered on BBM than on any other media (Appendix A). 

### 3.3. Comparison of ELF Communities among Five Individual Lichen Thalli 

Diversity of ELF communities was also influenced by individual lichen samples collected at different sites (Figure 6 and Appendix A). Total species richness detected in each lichen varied from 32 (thallus 4) to 45 (thallus 5). Seven species belonging to Sordariomycertes such as *Chaetomium globosum*, *Daldinia childiae* complex, *Hypoxylon perforatum*, Microascales sp. 1, *Nigrospora oryzae* complex, *Sarocladium kiliense*, and *Xylaria arbuscula* complex were commonly isolated from all the tested five lichen samples. On the other hand, 9, 13, 5, 9, and 18 species were unique for each individual lichen inhabited at five sampling sites, respectively (Figure 6).

## 4. Discussion

Our study showed that the diversity of the ELF communities can be affected by isolation conditions. Four variables were tested to optimize conditions for the isolation of diverse ELF communities from a foliose lichen. The middle part of thallus harbors more diverse ELF than the center and margin parts of thallus, but each part of thallus (different age of thallus) contained the unique ELF communities. The moderate sterilization of thallus surface can recover more diverse ELF communities without microbial contamination. Size of thallus fragments for inoculation seemed to be decisive to obtain most diverse ELF communities because smaller size can increase species richness by supporting slow-growing fungal recovery from a pool of ELF. Type of medium is also considered to be important factor for increasing species richness and isolate density. For example, slow-growing fungi were more frequently isolated on BBM than on any nutrient rich media.

Among four variables tested in this study, the sizes of thallus fragment was likely to be the most critical factor influencing species richness of ELF. In fact, use of small thallus fragment (1 mm^2^) covered all ELF taxa isolated from medium (25 mm^2^) and large (100 mm^2^) fragments. Previous studies normally used large fragments (100 or 25 mm^2^) of sterilized thallus for ELF isolation. The small fragment (1 mm^2^) firstly tested in this study was found to be highly efficient for the isolation of diverse ELF communities, compared to large (100 mm^2^) or medium (25 mm^2^) thallus fragments normally used in conventional isolation methods.

Several studies observed Sordariomycetes-dominant patterns in ELF communities [8,15,39,40,41]. The most abundant species, *Daldinia childiae* complex, was isolated from all individual lichen samples and under all isolation conditions. *D. childiae* was recovered both from the branch of pine trees and lichen thalli [42,43]. *D. childiae* has been reported to be a putative saprobe that stays in fresh host-plant tissue without causing the disease symptoms for a long time [44]. Therefore, endolichenic *D. childiae* in this study might have originated from the bark of pine trees, a substrate of host lichen *P. tinctorum*. *Sarocladium kiliense*, another abundant species, was also known as an endophytic fungus in maze and lichen [16,45]. Unlike *S. oryzae* causing sheath blast in rice, *S. kiliense* was not reported as a pathogen. *Trichoderma* sp. was the third most frequent ELF species in this study. Rhizocompetent filamentous fungal group, *Trichoderma* spp., were found in various environments comprised of soil, plants, and lichens [39,46,47]. Those species produce biocontrol agents against plant pathogens [47,48], but their enzymatic effects on lichen mycobiont are unclear. Therefore, Sordariomycetes species mentioned above are similar on their lifestyle to general endophytes and receive carbon inside the lichen thalli from the photobiont without causing pathogenic symptoms.

There were fungal species considered as specific or obligate occupants of the lichen thalli, e.g., lichenicolous fungi in previous studies [49,50]. The lichenicolous fungi were normally detectable only when they developed gall-like structures on lichen thalli. Lichenicolous fungus, *Cladophialophora* aff. *parmeliae* [51], was isolated in this study. Therefore, the modified method tested in this study recovered the known obligate fungal groups associated with the host lichen without the visible symptoms of gall-like structures [50]. Harutyunyan et al. had success in isolating of *Cladophialophora* sp. from healthy-looking thalli of *Protoparmeliopsis muralis* and *Physcia dimidiata* [52]. These findings agreed with the data obtained by metagenomic approach showing that several lichenicolous fungi might exist asymptomatically inside the lichen thallus [13].

Owing to the very slow growth rate of lichens, different parts of radially growing foliose lichen represent different ages of lichen thallus. Assuming that the growth rate of *P. tinctorum* is approximately 2 cm/year [53], the center, middle, and margin parts of the sample (thallus 3, for example) represent thalli of 6.25 (±1.25), 3.75 (±1.25), and 1.25 (±1.25) years old, respectively (Figure 1a). The ELF species of *Mollisia* sp., *Coniochaeta boothii*, and *Muscodor fengyangensis*, were uniquely isolated from the young marginal parts of the lichen thallus (Figure 3). *Mollisia* is a taxonomically neglected discomycete genus (*Helotiales*, Leotiomycetes) of saprotrophs commonly encountered on decaying plant tissues throughout the temperate regions [54]. *Mollisa* spp. were previously isolated from conifer needles. The endophytic *Mollisia* sp. may have alternatively colonized pine trees or the foliose lichen growing on the bark of the same pine tree. *Coniochaeta* species were present in several habitats such as plants, excrement, humus, and lichens [55,56]. *Muscodor* species are well-known endophytes that produce compounds lethal to pathogenic bacteria and fungi [57,58]. *C. boothii* and *M. fengyangensis* were reported as ELF for the first time. 

In contrast to the young marginal part, the oldest central part of lichen thallus would harbor not only fungal species colonized from the early stage of the lichenization but also fungal groups recently colonized the thalli. Unlike higher plants, lichens lack a cuticle as a protective layer. To take water up from their surfaces, lichens are necessarily open systems, and for the same reason, most lichen-inhabiting fungi are not confined to the part of lichen thallus. Several species of *Annulohypoxylon*, *Biscogniauxia*, *Hypoxylon*, and *Nigrospora* were recovered from center, middle, and margin parts of the lichen thallus (Figure 3). These cosmopolitan genera are known as endophytes or facultative saprotrophs. Although the highest species richness of ELF was recorded in the middle part of the lichen thallus, the marginal part and central part still harbored their own unique ELF diversity of 6 and 7 species, respectively (Figure 5b). Therefore, several fragments of the whole lichen thallus from the center to the margins of foliose lichens should be used to isolate diverse ELF which can be undiscovered fungal taxa and may be potential novel bioresources for pharmacological purposes.

Because mild sterilization incompletely removed fungal contaminants from the surface of the lichen thallus, it was unsuitable for the isolation of ELF. Epiphytic fungal species isolated after mild sterilization of the lichen thallus might be regarded as saprobes, including the genera *Annuloxypoxylon*, *Hypoxylon*, and *Muscodor* (Figure 3) [59,60]. In contrast, severe sterilization significantly reduced the diversity of ELF, and possibly damaged the fungal tissues at the medulla underneath of thallus cortex. A lichenicolous species *Cladophialophora* aff. *parmeliae* was isolated under our stringent sterilization condition. This fungus was melanized in axenic cultures as other *Cladophialophora* species were [61]. This melanin accumulation may protect the fungus from the sterilization agent. *Virgaria nigra* was also recovered from the thallus fragments treated with severe surface sterilization (Figure 3). *Virgaria* spp. have been reported to be soil-borne and endophytes [62,63]. *V. nigra* was previously isolated from *Peltigera dilacerata* after sodium hypochlorite or hydrogen peroxide sterilization, but not after ethanol sterilization [16]. Moderate sterilization completely removed fungal contaminants from the thallus surface and recovered the highest species richness of the ELF. This suggests that moderate sterilization (70% ethanol following 0.4% NaOCl, each for 90 s) can be applicable to isolating diverse ELF without interference from fungal saprobes on the thallus surface.

Previous studies normally used large fragments (100 or 25 mm^2^) of sterilized thallus for ELF isolation. In our study, smaller fragments (1 mm^2^) were found to be more efficient for the isolation of diverse ELF by supporting slow-growing ELF recovery. When the large- and medium-sized thallus inocula (100 and 25 mm^2^ in this study) were tested, mainly fast-growing ELF species were detected in a few days. Once fast-growing fungi began to cover the surface of the medium, it was difficult to isolate slow-growing fungi [64]. The diversity of ELF species appeared to be negatively related to the size of thallus fragment. Slow-growing rhytismataceous fungi were difficult to isolate when large leaf disks were used in a previous study [65]. The use of tiny pieces of thallus inoculum (1 mm^2^) allowed to isolate more diverse ELF communities. As a result, slow-growing fungal taxa such as *Pithya* aff. *vulgaris*, *Anteaglonium* sp., *Teichospora* sp., Dothideomycetes sp., *Cladophialophora* aff. *parmeliae*, Tympanidaceae sp., *Mollisia* sp., and Microascales spp. were uniquely isolated from the small pieces of thallus. Apothecia of the cup fungus, *Pithya* spp. can be found on the bark or rock [66]. The species from the rare genus *Anteaglonium* are associated with decorticated wood [67,68]. *Pithya* aff. *vulgaris* and *Anteaglonium* sp. were first reported as endolichenic fungi of corticolous lichen.

The highest species richness of ELF was obtained on PDA, a common medium that supports growth of a variety of fungi [69]. The use of enrichment media, however, would allow fast-growing fungi to overgrow [70]. Thus, the use of at least two types of medium, enrichment medium for the growth of general fast-growing fungi and minimal medium for the isolation of slow-growing fungi, might ensure the isolation of fungal strains with different growth rates. The BBM was designed originally for the cultivation of green algae [71]. This carbon-deficient medium made it possible to isolate slow-growing fungi, including *Cladophialophora* aff. *parmeliae*, *Teichospora* sp., and *Tympanidaceae* sp. (Figure 3). Accordingly, *Cladophialophora bantiana* was previously isolated using sterilized mineral oil without a carbon source [72]. Therefore, the use of different nutrient media, such as BBM and PDA, would ensure the recovery of the broad spectrum of ELF from the lichen thalli.

## 5. Conclusions

The following conditions were applicable to isolating more diverse ELF: (1) use of whole thallus from the center to the margin of foliose lichens, (2) removal of surface fungal contaminants through the moderate sterilization of lichen thallus (70% ethanol followed by 0.4% NaOCl, each for 90 s), (3) inoculation of tiny pieces (1 mm^2^) of sterilized thallus, and (4) the incubation of tiny pieces on two different types of nutrient medium: PDA (enriched medium) and BBM (minimal medium). Among the tested isolation conditions, the use of small fragments and different cultivation media apparently improved the efficiency of isolation of diverse and unexplored ELF from the lichen thallus. Diversity of ELF communities was also influenced by individual lichen samples collected at different sampling sites in our study.

## Figures and Tables

**Figure 1 jof-07-00335-f001:**
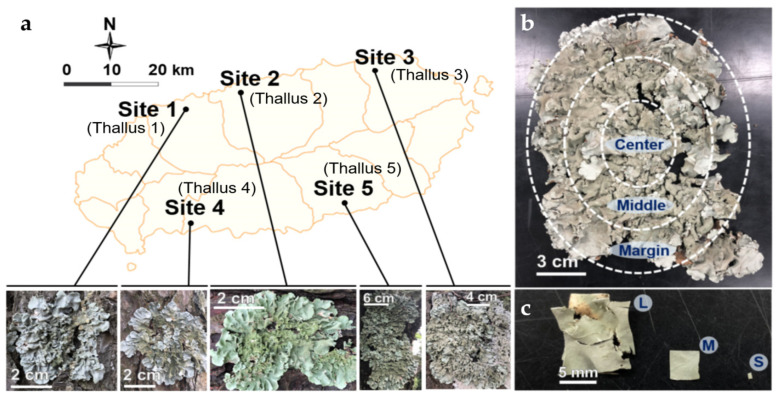
Collection sites and preparation of the thallus samples for ELF isolation. (**a**) *Parmotrema tinctorum* were collected from five sites in Jeju Island, South Korea; (**b**) Lichen thallus was divided into three parts by its diameter (center, middle and margin); (**c**) The piece of the lichen thallus was cut into three different sizes of inoculum (large: 100 mm^2^; medium: 25 mm^2^ and small: 1 mm^2^).

**Figure 2 jof-07-00335-f002:**
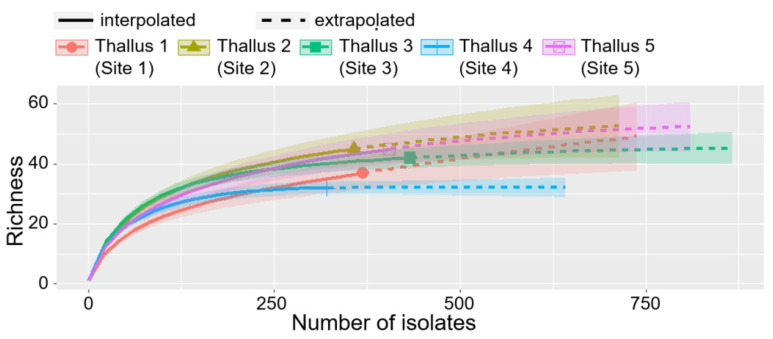
Rarefaction curves with 95% confidence intervals of estimated species richness of isolated ELF community.

**Figure 3 jof-07-00335-f003:**
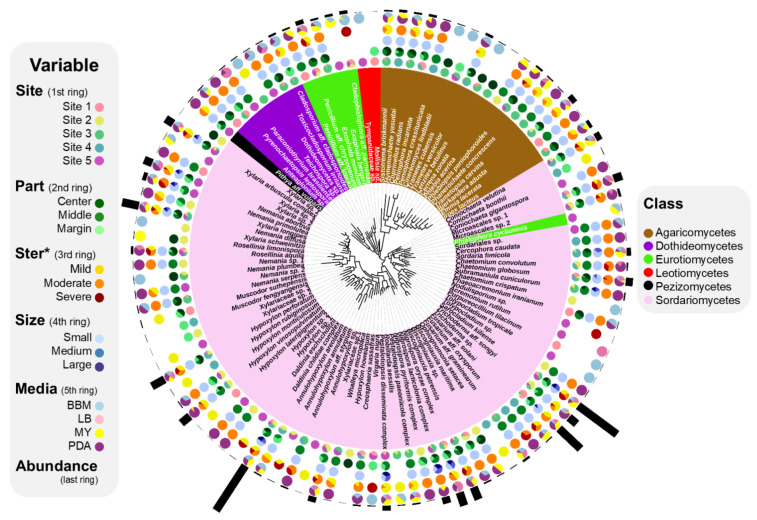
Phylogenetic relationships of the isolated ELF and relative abundances of classes in ELF communities isolated from different lichen thalli, different parts of lichen thallus, after different regimes of thallus surface sterilization, from fragments of different sizes and on different types of medium. Ster*, Sterilization.

**Figure 4 jof-07-00335-f004:**
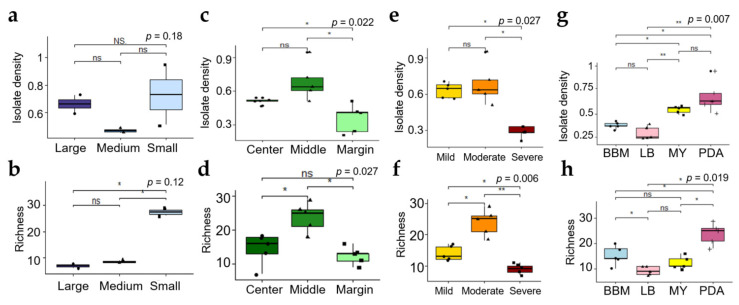
Isolate density and species richness of ELF communities from thallus fragments of different sizes (**a**,**b**), from different thallus parts (**c**,**d**), after different sterilization regimes (**e**,**f**), and on different media (**g**,**h**); Data of box plots are mean ± s.d. (*n* = 5, except for (**a**,**b**); *n* = 2); ** *p* < 0.01, * *p* < 0.05.

**Figure 5 jof-07-00335-f005:**
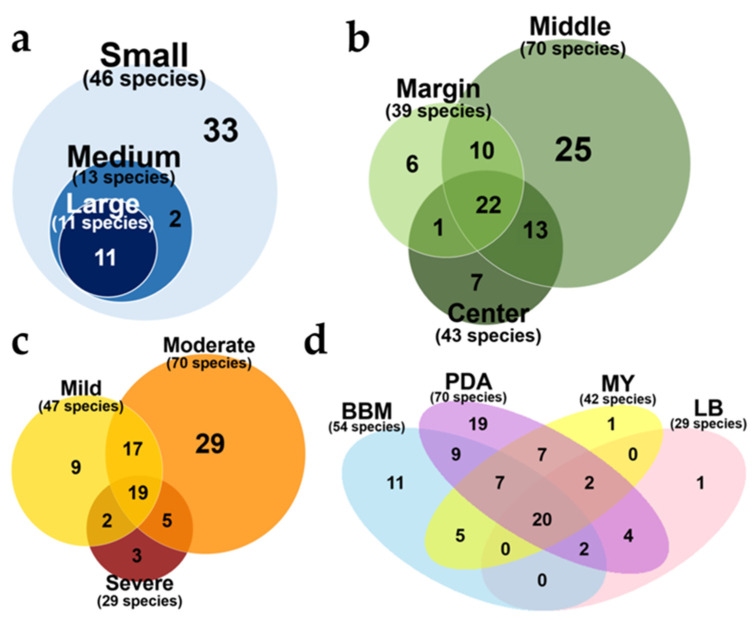
Venn diagrams showing relation among ELF communities from thallus fragments of different sizes (**a**), from different thallus parts (**b**), after different sterilization (**c**), and on different media (**d**).

**Figure 6 jof-07-00335-f006:**
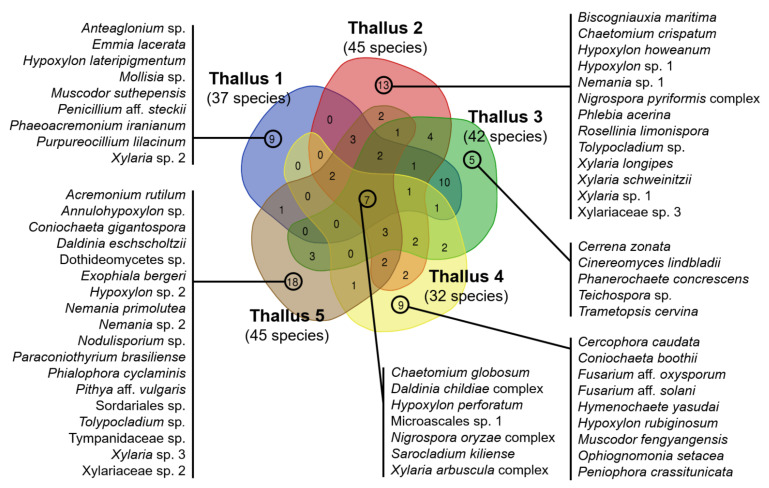
Composition of ELF communities isolated from five individual lichen thalli inhabited at 5 sampling sites. Lists of ELF from five individual lichen thalli represent unique species isolated from each lichen sample at each site.

## Data Availability

The datasets are included within the article and Appendix A. Nucleotide sequences reported in this article are available via GenBank.

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
