# Peer review of "Effect of Isolation Conditions on Diversity of Endolichenic Fungal Communities from a Foliose Lichen, Parmotrema tinctorum"

_jof, 2021, doi:10.3390/jof7050335_

Round 1
Reviewer 1 Report
The document is well writing and data are clearly presented and is suitable for publication in Journal of fungi with minor revisions.
The authors should pay attention to the following points
- In the abstract, rephrase “Moreovers, use of the small thallus….in the study” to be clearer
- In the discussion section, rephrase “the middle part of the thallus….. contained the unique ELF groups”
Author Response
We revised the MS using track-changes in Word program and provided point-by-point responses to the reviewers’ comments and concerns. I hope the revised MS can be acceptable for publication in Journal of Fungi.
Comment: In the abstract, rephrase “Moreovers, use of the small thallus….in the study” to be clearer
Response: modified as below
Moreover, ELF species isolated from the small thallus fragments covered all ELF taxa detected from the medium and the large fragments in this study.
Comment: In the discussion section, rephrase “the middle part of the thallus….. contained the unique ELF groups”
Response: modified as below
The middle part of thallus harbors more diverse ELF than the center and margin parts of thallus, but each part of thallus (different age of thallus) contained the unique ELF communities
Reviewer 2 Report
The manuscript “Effect of Isolation Conditions on Diversity of Endolichenic Fungal Communities from a Foliose Lichen, Parmotrema tinctorum” by Yang and col., provides interesting results concerning the influence of different procedures on isolation of ELF from foliose thalli. This is a relevant and interesting topic, and the presented work seems to me useful to improve the knowledge about these protocols; however, I feel the authors need to make some minor changes.
In particular the sampling size is a critical point: five thalli are really a small sample and one thallus for each sampling site is enormously small sampling size. Although the number of tests and replicates carried out in the laboratory for each thesis is sufficient for understanding the effects of 4 tested variables an inference of effects on diversity of ELF due to sampling site is not allowed with a single sample for each site. Authors cannot say if the differences on EFL isolated are due to a site-factor or to a stochastic one. With a one sample for each site authors may have been detected “individual” differences (if we can use the concept of “individual” for a lichen thallus). For this reason I suggest strongly rewriting the 3.3 paragraph. Results can be discussed in terms of “ELF diversity in different thally” not as “ELF diversity in different sites”. If authors desire to discuss the “site” effects they must provide an adequate sampling size (several thalli for each sampling site).
Some minor remark are listed below:
Abstract:
- “Several variables were modified in this study” = maybe “The influence of four variables was investigated” is more clear
- “foliose lichen, Parmotrema tinctorum, on pine trees.” in the abstract in irrelevant (and also a bit confusing) from which substrate the thalli have been collected.
- “Species composition and richness of ELF communities from different sites differed from each other in this study.” = data does not support this statement cause the poor number of specimens!
- Introduction
- “Muggia et al. reported the effects of different media on the composition of isolated ELF” = maybe “reported the effects of different media composition on the diversity of isolated EFL” ?
- -(thallus age, severity of surface sterilization, inoculum size, and types of medium) = maybe “cultural medium composition”?
- Materials and Methods
2.1
- for the following reasons. = “:” instead of “.” and also for the following phrases, I think it is better not to divide a list with full stops (but with semicolons) and avoid capital letters: it is clearer for the reader.
- the description of five sites can be shortened…
2.2
- “Modification of the isolation conditions was attempted as follows.” = The description is a bit confused. Can authors provide a sort of graphic diagram (in supplementary materials) illustrating exactly all the passages for each sample?
- The lichen thallus was divided into three parts = maybe “Each lichen thallus” in clearer
- center, middle, and margin = margin or edge?
2.4
- Constructing the Venn diagrams was performed using the VennDiagram package = maybe “the construction of the Venn diagrams”
- Results
- In the figure 2 is better labelled as “thallus1”/”thallus5” than “Site1”/”site5” and a similar change must be applied in the entire section “Results”
3.3. Comparison of ELF communities among sampling sites = this paragraphs should be deeply modified
- Discussion
Also in this section geographical discussions should be eliminated or deeply mitigated.
Author Response
Dear Editor
We are grateful to the reviewers for their insightful comments and constructive syuggestions. We have revised our manuscript with care in response to reviewer’s comments and incorporated the suggestions into our revised manuscript. We revised the MS using track-changes in Word program and provided point-by-point responses to the reviewers’ comments and concerns. I hope the revised MS can be acceptable for publication in Journal of Fungi.
Responsed to reviewer 2
Main comment: the sampling size is a critical point:
Authors cannot say if the differences on EFL isolated are due to a site-factor or to a stochastic one. With a one sample for each site authors may have been detected “individual” differences (if we can use the concept of “individual” for a lichen thallus). For this reason I suggest strongly rewriting the 3.3 paragraph. Results can be discussed in terms of “ELF diversity in different thally” not as “ELF diversity in different sites”.
Response: Thank you for your inshightful comment and concern. I agreed with your argue and revised the MS according to your comments.
- We changed “different sites” to “individual lichen thalli/samples” or “different lichen thalli” through the whole text and figures and lengends.
- We rewrote the 3.3. paragraph as below.
3.3. Comparison of ELF communities among five individual lichen thalli
Diversity of ELF communities was also influenced by individual lichen samples collected at different sites (Figure 6, Figure S3). Total species richness detected in each lichen varied from 32 (thallus 4 ) to 45 (thallus 5). Seven species belonging to Sordariomycertes such as Chaetomium globosum, Daldinia childiae complex, Hypoxylon perforatum, Microascales sp. 1, Nigrospora oryzae complex, Sarocladium kiliense, and Xylaria arbuscula complex were commonly isolated from all the tested five lichen samples. On the other hands, 9, 13, 5, 9, and 18 species were unique for each individual lichen inhabited at five sampling sites, respectively (Figure 6).
- We removed “site effect (different sites) on ELF diversity” in result and discussion. Instead, we used “differnt thalli or individual lichen samples” to discuss ELF diversity.
Minor remarks
Comment: “Several variables were modified in this study” = maybe “The influence of four variables was investigated” is more clear.
Response: modified as below
The influence of four variables were investigated in this study to determine the suitable conditions ......
Comment: “foliose lichen, Parmotrematinctorum, on pine trees.” in the abstract in irrelevant (and also a bit confusing) from which substrate the thalli have been collected.
Response: deleted “on pine tree” as below
for the isolation of more diverse ELF from a radially growing foliose lichen, Parmotrema tinctorum.
Comment: “Species composition and richness of ELF communities from different sites differed from each other in this study.” = data does not support this statement cause the poor number of specimens!
Response: modified as below
Species composition and richness of ELF communities from different lichen thalli differed from each other in this study.
Comment: “Muggia et al. reported the effects of different media on the composition of isolated ELF” = maybe “reported the effects of different media composition on the diversity of isolated EFL” ?
Response: modified as below
Muggia et al. reported the effects of different media on the community of isolated ELF.
Comment: -(thallus age, severity of surface sterilization, inoculum size, and types of medium) = maybe “cultural medium composition”?
Response: modified as below
four isolation variables such as thallus age, severity of surface sterilization, inoculum size, and types of medium were tested to optimize isolation conditions
Comment: for the following reasons. = “:” instead of “.” and also for the following phrases, I think it is better not to divide a list with full stops (but with semicolons) and avoid capital letters: it is clearer for the reader.
Response: modified as below
for the following reasons: it is distributed widely in the tropical and subtropical regions of the world [17, 18]: it is loosely attached to the substrate so that the thallus can be peeled off easily and cleanly from the substrate, e.g., bark and rock: mature thallus of P. tinctorum has sufficient biomass for the test.
Comment: the description of five sites can be shortened…
Response: modified as below
Five collection sites were the followings: site 1 (33�28′31″ N, 126�21′09″ E) is located at seaside cliff and directly exposed to strong wind from the sea: site 2 (33�30′28″ N, 126�28′01″ E) is a hill near the sea: site 3 (33�32′59″ N, 126�45′26″ E) is located at the top of the rising small defunct volcano which has been used as a graveyard and surrounded by agricultural field (citrus farms): site 4 ( 33�14′1″ N, 126�22′59″ E) is located at a small park nearby the sea: site 5 (33�16′23″ N, 126�42′14″ E) is located along the seaside with densely-developed viridian forest (Figure 1a).
Comment: “Modification of the isolation conditions was attempted as follows.” = The description is a bit confused. Can authors provide a sort of graphic diagram (in supplementary materials) illustrating exactly all the passages for each sample?
Response: Yes, we provided graphic diagram in supplementary materials.
Comment: The lichen thallus was divided into three parts = maybe “Each lichen thallus” in clearer
Response: modified as below
Each lichen thallus was divided into three parts
Comment: Constructing the Venn diagrams was performed using the VennDiagram package = maybe “the construction of the Venn diagrams”
Response: modified as below
Construction of the Venn diagrams was performed using the VennDiagram package
Comment: In the figure 2 is better labelled as “thallus1”/”thallus5” than “Site1”/”site5” and a similar change must be applied in the entire section “Results”
Response: We changed “site” to “thallus” in all figures and text.
Comment: Comparison of ELF communities among sampling sites = this paragraphs should be deeply modified
Response: We rewrote the 3.3 paragraphs as described above (page 2)
Comment: Discussion Also in this section geographical discussions should be eliminated or deeply mitigated.
Response: We deleted the related sentences in discussion.
Round 2
Reviewer 1 Report
The manuscript has been modified as suggested and is now suitable for publication.
Reviewer 2 Report
This new revised version of the manuscript by Yang and co-workers have been effectively improved, and I think that this new version is suitable for publication in JoF. Only two remarks:
- authors should correct as below the punctuation of this sentence (my previous comment was not clear enough):
"for the following reasons: it is distributed widely in the tropical and subtropical regions of the world [17, 18]; it is loosely attached to the substrate so that the thallus can be peeled off easily and cleanly from the substrate, e.g., bark and rock; mature thallus of P. tinctorum has sufficient biomass for the test."
- Also the caption of Figure S1 (very informative, great job!) may be modified as follow:
“Figure S1. Schematic diagram of experimental design adopted for ELF isolation.”